# Removal of Various Hazardous Materials Using a Multifunctional Biomass-Derived Hydroxyapatite (HAP) Catalyst and Its Antibacterial Effects

**Sanha Jang [1], Kyeongmun Park [1], Sehwan Song [2], Haksoo Lee [3], Sungkyun Park [2], Buhyun Youn [3],\* and Kanghyun Park [1,4],\***

[1] Department of Chemistry, Pusan National University, Busandaehak-ro 63beon-gil, Geumjeong-gu, Busan 46241, Korea; jangs0522@naver.com (S.J.); pkmkm951@naver.com (K.P.)

[2] Department of Physics, Pusan National University, Busandaehak-ro 63beon-gil, Geumjeong-gu, Busan 46241, Korea; sehwansong465@gmail.com (S.S.); pak@pusan.ac.kr (S.P.)

[3] Department of Biological Sciences, Pusan National University, Busandaehak-ro 63beon-gil, Geumjeong-gu, Busan 46241, Korea; hs950303@gmail.com

[4] SoulDot Co., Ltd., Pusan National University, Busandaehak-ro 63beon-gil, Geumjeong-gu, Busan 46241, Korea

\* Correspondence: bhyoun72@pusan.ac.kr (B.Y.); chemistry@pusan.ac.kr (K.P.); Tel.: +82-51-510-2264 (B.Y.); +82-51-510-2238 (K.P.)

**Abstract:** In the present study, oyster shells, a cause of environmental pollution, were employed effectively to synthesize hydroxyapatite (HAP) by facile oxidation and phosphorylation. The ability of HAP to adsorb various metal cations and inhibit bacterial growth was validated. The biomass-derived HAP catalyst exhibited high metal cation adsorption in water at room temperature and under various acidic conditions (M = Cr, Mn, Ni, Cu, Cd, Ba, and Pb). HAP was demonstrated to have a maximum removal efficiency of 92.8% for the heavy metal Pb. Even under different pH conditions, HAP was demonstrated to be effective for the removal of three harmful heavy metals, Cr, Cd, and Pb, with a particularly high removal efficiency demonstrated for Pb under all conditions (average removal efficiency of Cr: 63.0%, Cd: 59.9%, and Pb: 91.6%). In addition, HAP had a significant influence on phosphate ion adsorption in aqueous solution, eliminating 98.1% after 3 min. Furthermore, biomass-derived HAP was demonstrated to have significant antibacterial activity against *E. coli* and *S. aureus* (5 mM: 74% and 78.1%, 10 mM: 89.6% and 96.0%, respectively).

**Keywords:** biomass-derived material; nanomaterial; environmental chemistry; multifunctional catalyst; antibacterial; metal ion adsorption; phosphate ion adsorption

## 1. Introduction

Environmental pollution, especially in the ocean and groundwater, is increasing rapidly because of the organic and inorganic pollutants in hazardous wastewater [1–10]. These pollutants come from industrial wastewater and electrical devices containing heavy metals, such as Pb, Cd, and Cr. Pb, in particular, is a hazardous metal that causes osteoporosis, and Cd is a hazardous metal that causes itai-itai disease [2,11–13]. The elimination of heavy metals is of particular interest to scientists and engineers because of the negative impact of these pollutants on a wide range of living organisms [14,15]. As a result, various techniques for heavy metal removal from wastewater, such as ion exchange, chemical precipitation, membrane separation, and adsorption, have been developed [16,17]. Precipitation accompanied by a chemically or externally mediated transition to non-labile forms is a popular method for removing metals from water and wastewater. The long-lasting matrix to which the metals are transferred must be stable under various conditions, especially pH and temperature [11]. Auaok et al. reported the effectiveness of Pb (II) adsorption using polypyrrole-based activated carbon [18]. Li et al. was successful in adsorbing heavy metals through zeolite, a material with a large surface area and pores [17].

In an earlier study, we successfully used $Sb_2S_5$ nanofibers to selectively adsorb Pb in water at low temperatures [19]. According to the literature, heavy metals such as Pb, Ni, Cu, and Cd, can be quickly eliminated from wastewater using hydroxyapatite (HAP, $Ca_{10}(PO_4)_6(OH)_2$) [20–22].

Phosphorus (P) is an important nutrient for all species and is required for the healthy operation of ecosystems [23]. Phosphoric acid ($H_3PO_4$) is decomposed in the aqueous phase at various $pK_a$ values to yield phosphate species ($H_2PO_4^-$ at $pK_a$ = 2.15, $HPO_4^{2-}$ at $pK_a$ = 7.20, and $PO_4^{3-}$ at $pK_a$ = 12.35). Thus, the total phosphate is the sum of all anionic phosphates [24]. However, excess P that is not digested in the biogeochemical P cycle is released into the waterways, causing eutrophication [25]. In aquatic environments, eutrophication causes toxic algal blooms, biodiversity loss, and dead zones [26]. As a result, effective and practical techniques to remove phosphate from water bodies are critical [27]. HAP has been used to confirm the replacement of phosphate anions in previous studies [21,28,29].

Microbial toxins such as total coliform, Escherichia coli (*E. coli*), fecal coliform, and Salmonella are often found in groundwater where treatment facilities are lacking. This microbial contamination results in acute and chronic diseases, such as diarrhea and high fever [30–32]. The high volume of sewage dumped into water bodies is a major issue in the industrial era [1]. Significant research efforts have been aimed at developing a catalyst that utilizes chemical and physical adsorption to solve this problem. In addition, many engineers and scientists have worked tirelessly to eliminate bacteria. Some materials, both organic and inorganic, have antibacterial properties [33–40]. Lamkhao et al. recently reported that a microwave-assisted combustion process can be used to produce HAP with antibacterial properties [38].

Consumption of seafood produced by the aquaculture sector has increased in recent years as the world's population has grown, and interest in the recycling of industrial waste from seashells has also grown [41–43]. According to data from the National Assembly Research Service in South Korea [44], approximately 40% of fish and shellfish output was discarded each year between 2010 and October 2019 (Figure S1). Seashells have been employed as raw materials for a variety of applications. Nakanishi [45], Shon [46], and Koopan [47] et al. reported on the recycling of oyster shells for various uses. Furthermore, Yeom et al. reported successful phosphate ion removal using a crab shell [48]. Oysters are a popular seafood worldwide. In particular, oyster shells made of calcium carbonate are aquaculture waste that causes serious disposal problems worldwide, as well as on the southeastern coast of South Korea (Figure S2). The oyster shells are easy to collect for recycling purposes and are therefore can be utilized as a calcium supply.

Herein, we report the preparation of HAP catalysts from oyster shells using facile methods such as heat treatment and phosphorylation, and characterize their metal cation adsorption efficiency and antibacterial performance. In this investigation, Pb, an extremely toxic heavy metal, was adsorbed at a rate of 92.8% by the HAP catalyst. Furthermore, as a result of experiments with various pH conditions that are representative of actual wastewater, a high removal efficiency was demonstrated for Pb, while Cr and Cd demonstrated maximum effectiveness at pH 7. When HAP was used to remove phosphate ions over a period of 9 min, the removal efficiency was found to be 98.1% at 3 min, and remained constant thereafter. Bacteria are one of the most dangerous organisms for humans; thus, we evaluated the antibacterial properties of *E. coli* and *S. aureus*. We validated the antibacterial capabilities of HAP against *E. coli* and *S. aureus* (In the case of 5 mM of HAP, *E. coli*: 74.4% and *S. aureus*: 74.9%. in the case of 10 mM of HAP, *E. coli*: 90.9% and *S. aureus*: 96%, respectively).

## 2. Results and Discussion

### 2.1. Preparation and Characterization of HAP Catalyst

CaO was produced from oyster shells according to Equation (1) using a simple heat treatment at 900 °C. We used existing research as a guide and synthesized the HAP catalyst according to Equation (2) [45].

$$CaCO_3 \rightarrow CaO + CO_2 \tag{1}$$

$$10CaO + 6H_3PO_4 \rightarrow Ca_{10}(PO_4)_6(OH)_2 + 8H_2O \tag{2}$$

Transmission electron microscopy (TEM) images revealed the calcium oxide formed by oxidation, and the HAP formed by phosphorylation, in the initial calcium carbonate oyster shells (Figure 1a,c). There was no discernible form to the CaO, but there was an indication of a definite rod shape morphology for the HAP. This was caused by the phosphorylation of CaO. The high-resolution TEM (HRTEM) images and Fourier-transform patterns revealed the formation of CaO crystals with a lattice distance of 0.242 nm and HAP crystals with a lattice distance of 0.280 nm, which correspond to the (200) and (211) planes, respectively (Figure 1b,d).

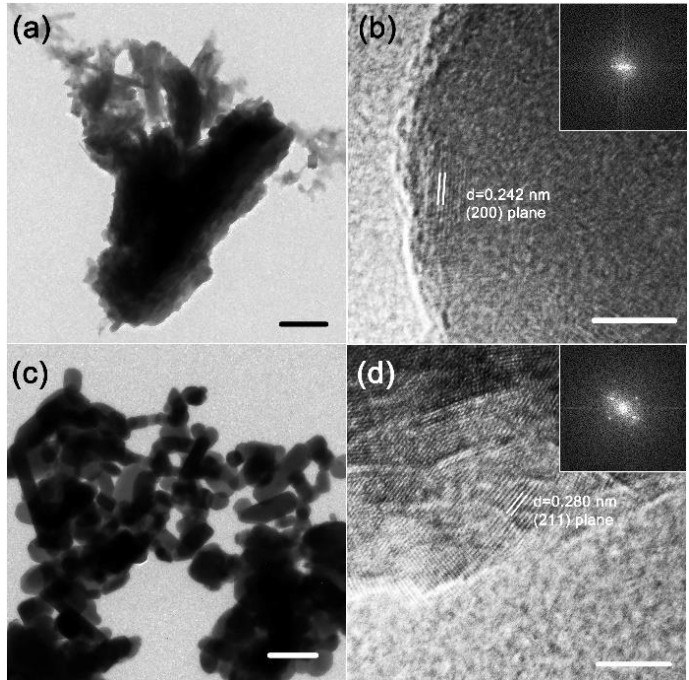

**Figure 1.** (**a**) TEM and (**b**) HRTEM (inset: SAED pattern) images of CaO. (**c**) TEM and (**d**) HRTEM (inset: SAED pattern) images of hydroxyapatite (HAP). The bars indicate 500 nm (**a**,**c**) and 5 nm (**b**,**d**), respectively.

Figure 2 illustrates the element mapping analysis obtained from high-angle annular dark-field scanning transmission electron microscopy (HAADF-STEM) images. The HAADF image exhibited bright particles (Figure 2a). The bright particles of HAP in the HAADF image are more angular than the brilliant particles of CaO in the CaO HAADF image (Figure S3a). Ca (green), P (cyan), and O (red) elemental maps depict the distribution of Ca, P, and O, which are components of HAP (Figure 2b–d). The existence of P was not verified in the elemental analysis of CaO, while Ca (green) and O (red) were observed, with the strongest signal corresponding to Ca (Figure S3b–d).

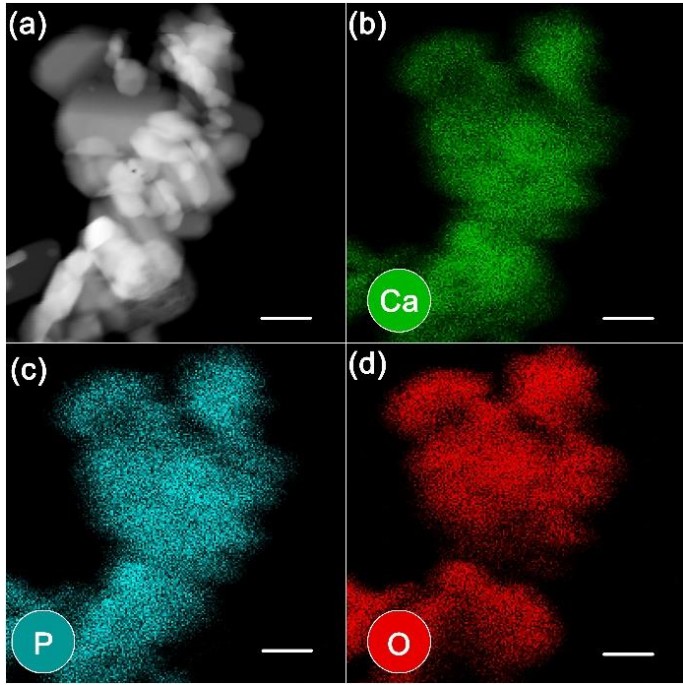

**Figure 2.** (**a**) HAADF-STEM image and (**b**–**d**) elemental mapping images of HAP. The bars indicate 100 nm.

Figure 3a shows the Fourier-transform infrared (FT-IR) spectra of CaO and HAP. The broad and sharp peaks at 1383–1546 cm$^{-1}$ and 875 cm$^{-1}$ are characteristic CaO peaks. The chemical groups $PO_4^{3-}$, $OH^-$, and $HPO_4^{2-}$ that are typical of HAP were observed in the FT-IR spectrum. The $PO_4^{3-}$ group provides a significant IR transmission band at 568 and 602 cm$^{-1}$ and 1000–1100 cm$^{-1}$. The transmission wavelength of water causes distinct peaks at 3572 cm$^{-1}$ and 633 cm$^{-1}$. In the XRD pattern, the CaO peak was observed at 2θ = 32.3°, 37.5°, 54.0°, 64.4°, and 67.6°, corresponding to the reflections of the (111), (200), (220), (311), and (222) planes of CaO, respectively (JCPDS No. 82-1691) (Figure 3b). However, when the oyster shell was oxidized, it was not entirely transformed into calcium oxide, and some calcium carbonate remained. After the phosphorylation of HAP, there were numerous peaks other than that of CaO, with particularly prominent peaks appearing at 31.7°, 32.2°, and 32.9°, corresponding to the reflections of the (211), (112), and (300) planes of HAP, respectively (JCPDS No. 09-0432). According to the literature [20], CaO is not entirely phosphorylated, and therefore the results represent a variety of different crystal structures for CaO. As a result, among the HAP XRD peaks, there is a peak at 37.5°. The lattice distance determined by HRTEM was quite consistent with the information interpreted from the XRD peak.

The chemical states of the samples were examined using XPS. The survey scan of CaO and HAP, shown in Figure S4a,b, demonstrates that all samples consisted of N, O, Ca, and C. In addition, P was found in the HAP powder. Figure 3c,d show the C 1*s* and P 2*p* core-level X-ray photoelectron spectra and deconvoluted results of CaO and HAP powders, respectively. All spectra were calibrated by adjusting the C 1*s* peak to 284.5 eV [49]. Furthermore, all the peaks were deconvoluted using a Gaussian-Lorentzian fitting using CASA. The C 1*s* spectra show surface carbon contamination (C-C, C-O-C, and O-C=O) and -CO$_3$ (carbonate) (Figure 3c). The relative -CO$_3$ peak, compared to C-C, is less dominant in the HAP powder due to the phosphorylation. Unfortunately, as there is only a slight difference in the binding energy between the source materials, CaCO$_3$ and HAP, it is difficult to separate the two peaks [50]. However, the lack of satellite loss feature, shown in Figure S4c in the Ca 2*p* spectra, provides an indication as to how HAP forms after the reaction [https://xpssimplified.com/elements/calcium.php accessed on 13 May 2021]. Therefore, the CaO powder is composed of CaCO$_3$, while HAP is

$Ca_{10}(PO_4)_6(OH)_2$. Furthermore, the P 2*p* spectrum is only represented in the HAP powder, as shown in Figure 3d. The O 1*s* spectra, shown in Figure S4d, also show additional HAP bonds and O-C=O, C-O-C, and $CaCO_3$ for the HAP powder. However, O-C=O, C-O-C, and $CaCO_3$ bonds were present in the CaO powder spectra.

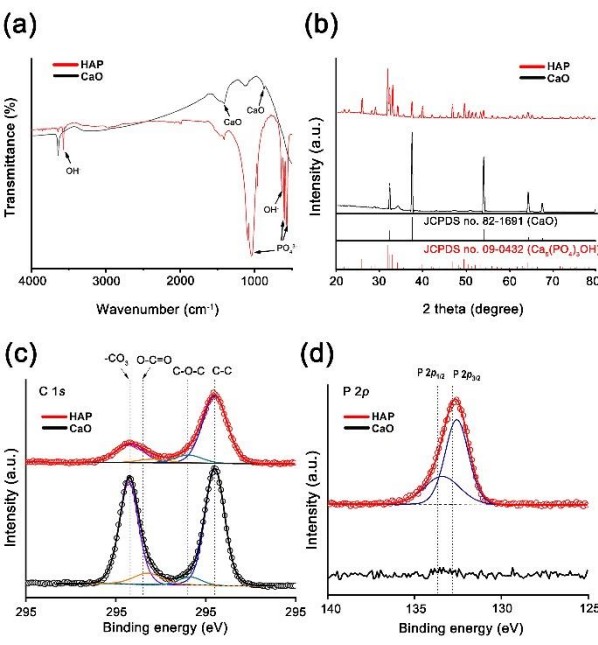

**Figure 3.** (**a**) FT-IR spectrum of CaO and HAP powder. (**b**) XRD data for CaO and HAP powder. Deconvoluted (**c**) C 1*s* and (**d**) P 2*p* X-ray photoelectron spectrum of CaO and HAP powder. The vertical dotted lines indicate the reference binding energy [50,51].

### 2.2. Metal Cation Removal Efficiecny of HAP Catalyst

To test the catalytic performance of HAP as an adsorbent for metal cations in water, we prepared a model solution containing seven metal ions, $Cr^{3+}$, $Mn^{2+}$, $Ni^{2+}$, $Cu^{2+}$, $Cd^{2+}$, $Ba^{2+}$, and $Pb^{2+}$. To prepare 1 L of 100 ppm metal cation solution, known quantities of metal nitrates were mixed with distilled water. In a typical removal test, biomass-derived HAP catalyst (0.01 mmol) was inserted into the model solution (50 mL), and the HAP catalyst was collected by centrifugation after 24 h of stirring at RT. In addition, we created a pH 6 solution to represent wastewater using $HNO_3$ (0.1 M) and NaOH solution (0.1 M). Inductively coupled plasma-optical emission spectroscopy (ICP-OES) was used to evaluate the residual aqueous solutions. Through ICP-OES analysis, we discovered that $Cr^{3+}$, $Mn^{2+}$, $Ni^{2+}$, $Cu^{2+}$, $Cd^{2+}$, $Ba^{2+}$, and $Pb^{2+}$ were present at concentrations of 115.39, 85.44, 98.47, 112.60, 99.63, 114.79, and 103.52 ppm, respectively, in the model solution. Furthermore, we demonstrated the removal efficiency of the metal cations (Table S1).

The removal efficiency and the adsorption capacity ($q_e$) were calculated using Equations (3) and (4):

$$Re\% = \left( \frac{C_0 - C_e}{C_0} \right) \times 100 \tag{3}$$

$$q_e = \frac{C_0 - C_e}{W} \times V \tag{4}$$

where $C_0$ and $C_e$ (mg·L$^{-1}$) are the liquid-phase concentrations of the metal cations before and after adsorption, respectively. *V* (L) and *W* (g) are the volume and mass of each metal ion solution, respectively. Figure 4 illustrates the removal efficiencies of each metal cation. HAP was shown to be capable of removing each cation and a removal efficiency of more than 55% was demonstrated for the extremely harmful Cr, Cd, and Pb (Figure 4a). The elimination efficiency of Pb (92.8%), which causes osteoporosis, was especially high (Cr: 57.9% and Cd: 73.9%, respectively). Furthermore, HAP also exhibits Mn, Ni, Cu,

and Ba adsorption characteristics, and the removal rates were 66.3%, 37.1%, 29.1%, and 44.0%, respectively. Equation (3) was used to obtain the adsorption capacities. The HAP adsorption capacities were 66.8, 56.6, 36.6, 32.8, 73.6, 50.5, and 96.0 L·g$^{-1}$ of Cr, Mn, Ni, Cu, Cd, Ba, and Pb, respectively (Figure 4b). A high capacity for the adsorption of Pb, Cu, and Ni and several other metal cations was demonstrated for HAP. Numerous studies have established the potential of HAP to absorb metal ions via two major processes (Equations (5)–(8)). The first is ion exchange, which occurs when calcium ions in the HAP lattice are replaced by adsorbed metal ions via diffusion [52–57].

$$Ca_{10}(PO_4)_6(OH)_2 + xM^{2+} \rightarrow Ca_{10-x}M_x(PO_4)_6(OH)_2 + xCa^{2+} \tag{5}$$

$$Ca_{30}((PO_4)_6(OH)_2)_3 + 20Cr^{3+} \rightarrow Cr_{20}((PO_4)_6(OH)_2)_3 + 30Ca^{2+} \tag{6}$$

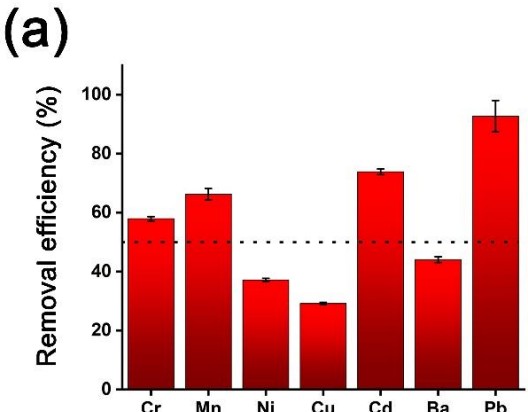

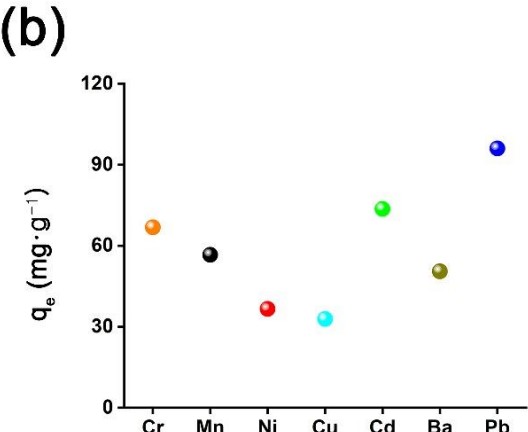

**Figure 4.** HAP catalytic performance at pH: (**a**) removal efficiency and (**b**) adsorption capacity. Reaction conditions: hydroxyapatite (0.05 g), pH 6, time 24 h, RT, respectively. The concentration of M ion aqueous solution exhibits in Table S1.

The other process involves the precipitation of metal phosphate complex materials in acidic environments. In this state, phosphate interacts with metallic ions according to the equations below, forming a new crystalline structure with limited solubility [52,58–65].

$$Ca_{10}(PO_4)_6(OH)_2 + 14H^+ \rightarrow 10Ca^{2+} + 6H_2PO_4^- + 2H_2O \tag{7}$$

$$10M^{2+} + 6H_2PO_4^- + 2H_2O \rightarrow M_{10}(PO_4)_6(OH)_2 + 14H^+ \tag{8}$$

According to Mobasherpour et al., adsorption is accomplished using a two-step adsorption technique [66]. Initially, Ni ions are rapidly adsorbed through surface interactions

with = POH sites, leading to the formation of $Ca_{10-x}M_x(PO_4)_6(OH)_2$. Thus, the initial process involves Ni adsorption by HAP, followed by $Ca^{2+}$ replacement by $Ni^{2+}$.

The pH of actual wastewater varies based on the surroundings. Thus, we selected, Cr, Cd, and Pb, which are known to be hazardous, to test under various pH conditions (Table S2). The pH of the solution has a considerable impact on heavy metal adsorption because it affects adsorbent characteristics such as surface charge, adsorbate speciation, and degree of ionization in aqueous solutions. Because each pH condition was simulated with solutions derived from the stock solution prepared in Figure 4, no further ICP-OES analysis of the stock solution was carried out. The same quantity of HAP was added to the stock solution, and a 24 h adsorption experiment was performed. Although the data from the experiments performed at pH 6 are already shown in Figure 4, they are also included in Figure 5 for comparison. Figure 5a–c depict the removal efficiency of heavy metal ions by the HAP catalyst at various pH levels. In general, the pH of wastewater is known to be 6 to 8; however, to account for the acidification produced by acid rain, the experiment was performed at pH 4, 5, 6, and 7. At pH 7, a good removal efficiency was demonstrated for the three heavy metal cations. $Cr^{3+}$ ions were removed more efficiently as the pH increased, whereas $Pb^{2+}$ ions were removed efficiently in a wide range of pH conditions. The removal efficiency for the Cd ion (15.6%) was lowest at pH 5, whereas the removal efficiency $Cr^{3+}$ was 44% at this pH level. According to the literature, because the surface of HAP is negatively charged in this range, the most appropriate pH range for $Cd^{2+}$ adsorption is above $pH_{pzc}$ (surface charge of HAP is zero). The pH can impact efficient ion exchange adsorption between cationic species and the $Ca^{2+}$ in HAP [67,68]. In contrast, HAP has a high adsorption capacity at pH 4, 5, 6, and 7 for Pb, which suggests that Pb ion adsorption in wastewater would be efficient (Figure 5d). Although the heavy metal removal efficiency varied with pH, the average removal efficiencies of Cr, Cd, and Pb were 63.0%, 59.9%, and 91.6%, respectively (Table S2).

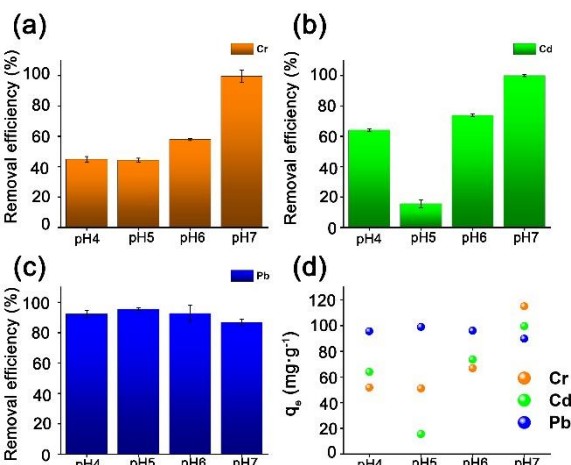

**Figure 5.** Removal efficiency using HAP under various pH conditions for three major hazardous metal cations: (**a**) Cr, (**b**) Cd, and (**c**) Pb. (**d**) Adsorption capacities. Reaction conditions: hydroxyapatite (0.05 g), pH 4, 5, 6, and 7. Time 24 h and RT, respectively. The concentration of M ion aqueous solution exhibits in Table S2.

### 2.3. Phosphate Ion Removal Efficiecny

According to the literature [69], phosphate ion removal in aqueous solution is caused by the flexibility of the structure surface activity and the non-stoichiometry of HAP. As a result, the phosphate ion adsorbs to the location next to the $Ca^{2+}$ ion on the HAP surface or to the phosphate ion defect. To explore the phosphate ion removal efficiency we applied the ascorbic acid solution method outlined in the experimental procedure with monitoring every 3 min (Figure 6). When performing UV-VIS analyses, we used a wavelength of 625 nm to account for the color reagent. We compared the phosphate ion removal efficiency

of HAP with that of CaO. The phosphate ion removal efficiency of HAP was demonstrated to be 98.1% after 3 min, and subsequent testing at 3 min intervals showed that the efficiency remained consistent thereafter. In contrast, CaO was demonstrated to have an ion removal efficiency of 71.1% after 3 min, at which point the solution was saturated. As a consequence, the phosphate ion removal efficiency of HAP was 27% greater than that of CaO, and saturation was achieved very rapidly.

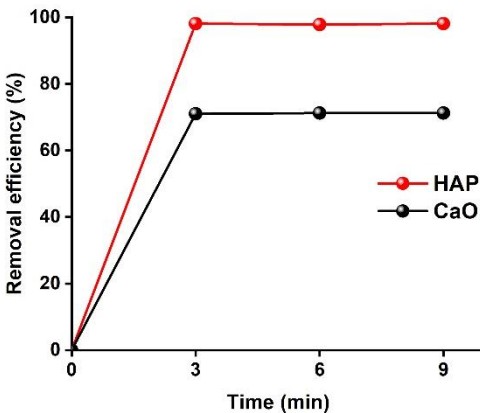

**Figure 6.** Comparison of phosphate ion removal efficiency using HAP and CAO. Reaction conditions: Hydroxyapatite (2 g) and CaO (2 g). The reaction time 0, 3, 6, and 9 and RT.

### 2.4. Antibacteiral Performance of HAP

The antibacterial performance of HAP (5 mM and 10 mM) was evaluated against Gram-negative and Gram-positive bacteria, *E. coli* and *S. aureus*. Further, 600 nm was chosen as an appropriate wavelength to measure the concentration of bacteria. Equation (9) was used to calculate the antibacterial efficiency.

$$\frac{I_0 - I}{I_0} \times 100 \tag{9}$$

$I_0$ and $I$ indicate the initial absorbance of the bacteria and the final absorbance, respectively. Before cultivating the bacteria, 100 mL of the HAP solution was added to the culture solution, and the bacteria were cultured for 24 h. Figure 7a shows the estimated antibacterial activity of HAP based on Equation (9). Furthermore, we demonstrated the antimicrobial efficiency of *E. coli* and *S. aureus* (Table S3). The antibacterial activity was 74.4% against *E. coli* and 74.9% against *S. aureus* when 5 mM HAP was used, indicating similar antibacterial effectiveness. Furthermore, when 10 mM HAP was applied, *E. coli* demonstrated 90.9% antibacterial effectiveness, while *S. aureus* demonstrated 96.0% antibacterial effectiveness. We used FE-SEM to examine the damaged bacterial cells (Figure 7b–e). In the case of *E. coli*, the cells appeared to be damaged or melted at 5 mM and 10 mM HAP, respectively (Figure 7b,c). Aggregation occurred with 5 mM HAP in *S. aureus*, and cell surface modification and aggregation occurred with 10 mM (Figure 7d,e). These results indicate that HAP demonstrates successful antibacterial performance against *E. coli* and *S. aureus*.

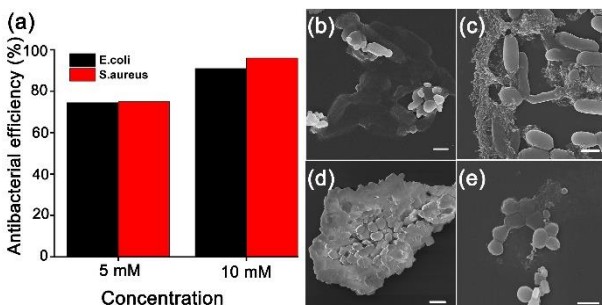

**Figure 7.** (**a**) Antibacterial efficiency using HAP. SEM images of antibacterial performance against *E. coli* (**b**) 5 mM and (**c**) 10 mM HAP. SEM images showing the antibacterial performance against *S. aureus* with (**d**) 5 mM and (**e**) 10 mM HAP. The bars indicate (**b**–**e**) 1 μm.

## 3. Materials and Methods

### 3.1. Chemicals and Characterization

Oyster shells were obtained from Tongyeong, South Korea. Solvents and all other chemicals were obtained from commercial sources and used as received, unless otherwise noted. Biomass-derived CaO and HAP catalysts were characterized using field-emission transmission electron microscopy (TALOS F200X), field emission scanning electron microscopy (FE-SEM), and elemental mapping (ZEISS SUPRA 25). X-ray diffraction (XRD) was performed on a Rigaku RINT 2200 HK diffractometer, and the chemical states were characterized using high-performance X-ray photoelectron spectroscopy (XPS). Fourier-transform infrared spectra (FT-IR) were recorded on a Spectrum Two spectrometer (PerkinElmer). Bacteria were characterized using UV-VIS (UV-1800, Shimadzu UV spectrophotometer).

### 3.2. Synthesis of Biomass-Derived HAP Catalyst from Oyster Shell

The discarded oyster shells were cleaned with clean water and dried in an oven at 140 °C for 24 h. The oyster shells (10 g) were mechanically ground in a mortar for 10 min under atmospheric conditions. To synthesize CaO powder, the powdered oyster shell was placed in an alumina bowl and heated at a ramping rate of 7.5 °C·min$^{-1}$ up to 900 °C under an air atmosphere, and then held at 900 °C for 4 h. After heating, the resulting yellowish-white powder was cooled to room temperature (RT). HAP powder was synthesized using the method reported by Nakanishi group [45]. To synthesize HAP powder, distilled water (30 mL), $H_3PO_4$ (30 mmol), and ammonia solution (10 mmol) were added to a 100 mL round-bottom flask and stirred for 10 min. Powdered CaO (2.804 g), $H_3PO_4$ (30 mmol), and $NH_4OH$ (10 mmol) were added to the round-bottom flask and stirred at 40 °C for 24 h. The subsequent colloidal dispersion was cooled to RT, washed three times with distilled water, and the sample was separated by centrifugation at 3500 rpm for 10 min. The precipitate was dried in an oven at approximately 100 °C. Finally, the powder was placed in an alumina bowl and heat-treated at a ramping rate of 6.7 °C·min$^{-1}$ up to 800 °C under an air atmosphere, and then held at 800 °C for 4 h.

### 3.3. Removal Efficiency of Various Metal Cation Species

First, we prepared various M nitrate salt solutions containing 100 ppm of M cations at RT (M = $Cr^{3+}$, $Mn^{2+}$, $Ni^{2+}$, $Cu^{2+}$, $Cd^{2+}$, $Ba^{2+}$, and $Pb^{2+}$). Experiments were carried out under two sets of conditions to determine the metal ion removal efficiency of the biomass-derived HAP catalyst.

1. Standard conditions: The biomass-derived HAP powder (50 mg) was dispersed in a metal ion solution (50 mL) and stirred at RT for 24 h. The pH of the solution was adjusted to 6 using nitric acid (0.1 M) and sodium hydroxide solution (0.1 M), to create a stock solution with conditions comparable to wastewater.

2.  pH controlled conditions: pH 4, pH 5, and pH 7 solutions were prepared to recreate severe and realistic wastewater conditions. Each solution was generated in the same manner as in Experiment 1.

### 3.4. Removal Efficiency for Phosphate Ion

We prepared a phosphate solution containing 100 ppm of $KH_2PO_4$ at RT. A volumetric flask (1 L) was filled with powdered $KH_2PO_4$ (4.393 g) and distilled water to make a phosphate solution. To test the phosphate ion adsorption capabilities of HAP, a pre-prepared phosphate solution (20 mL) was placed in a round-bottom flask, and after adding HAP (2 g), solution was collected every 3, 6, and 9 min, respectively. CaO was also tested under the same conditions as those used to test HAP. The phosphate was quantified using the ascorbic acid method, a color development reagent technique, and analyzed using UV-VIS [27]. Before creating a chromogenic reagent for phosphate analysis, three reagents were synthesized.

1.  Potassium antimonyl tartrate solution was mixed with potassium antimonyl(III) tartrate hydrate (6.8 mg) in distilled water (50 mL).
2.  An ammonium molybdate solution was prepared by combining ammonium molybdate tetrahydrate (1.5 g) with distilled water (50 mL).
3.  Ascorbic acid solution was prepared by combining ascorbic acid (0.88 g) with distilled water (50 mL).

To prepare the color reagent, sulfuric acid (14% *v/v*, 25 mL), potassium antimonyl tartrate solution (2.5 mL), ammonium molybdate solution (7.5 mL), and ascorbic acid solution (15 mL) were added to a 100 mL flask and stirred for 10 min. A phosphate solution (100 ppm of $PO^{3-}$) was prepared by diluting 1 mL of phosphate solution in a volumetric flask (10 mL) with water. Then, 5 mL of the diluted $H_3PO_4$ solution and 0.8 mL of pre-prepared color development reagent were combined, stirred, and measured after 5 to 10 min.

All of the above experiments were performed under standard conditions (RT and 1 bar).

### 3.5. Antimicrobial Test Using Gram-Netaive and Gram-Positive Bacteria

We evaluated the antibacterial effect by growing bacteria in different concentrations of HAP (Scheme S1). To determine the optimal concentration, we prepared 5 and 10 mM HAP in lysogeny broth (LB). Bacteria were incubated in an Erlenmeyer flask with LB (100 mL) and HAP at 37 °C for 24 h. We measured the bacterial concentration using UV-VIS at 600 nm to assess the efficacy of HAP for antibacterial activity. We used FE-SEM to detect the damaged bacterial cells.

### 4. Conclusions

In this study, we successfully synthesized biomass-derived HAP using low-value oyster shells, which are otherwise considered industrial waste. Due to the top-down approach of oxidation and phosphorylation, the purity of the HAP was low. However, it demonstrated high removal efficiency by metal cation adsorption, particularly for the Pb ion (92.8%). Generally, the pH of wastewater varies from environment to environment, and average removal efficiencies of 63.0% (Cr), 59.9% (Cd), and 91.6% (Pb) were demonstrated at pH 7, 5, and 4. Furthermore, because the phosphate ion removal efficiency (98.1%) was also high, the use of HAP could prevent freshwater and saltwater eutrophication. Moreover, as bacterial contamination is a component of water pollution, we confirmed the antibacterial properties of HAP. When we tested the antimicrobial activity against *E. coli* and *S. aureus* using 5 mM HAP, we achieved antibacterial efficacies of 74.4% and 78.1%, respectively. The antibacterial efficacies of 10 mM HAP were 89.6% and 96.0%, respectively. We expect that application of these biomass-derived products would reduce pollution and contribute to a cleaner environment for future generations.

**Supplementary Materials:** The Supplementary Materials are available online at https://www.mdpi.com/article/10.3390/w13223302/s1. Figure S1: Histogram of South Korean fishery production and byproducts [44]; Figure S2: Oyster shells that have not been treated in South Korea. The photo was taken by Sanha Jang; Figure S3: (a) HAADF-STEM image of CaO and (b-d) elemental mapping images of CaO. The bars indicate 100 nm, respectively; Figure S4: Survey spectrum of (a) CaO and (b) HAP powders and deconvoluted (c) Ca 2p and (d) O 1s spectrum. The vertical dotted lines indicate the reference binding energy; Table S1: ICP data table for RT condition of various metal cation adsorption; Table S2: ICP data table for various acidic condition of various metal cation adsorption; Table S3: Calculation of antibacterial efficiency of E. coli and S. aureus for 24 h. Scheme S1. Schematic image of antibacterial tests using Gram negative and Gram positive, respectively.

**Author Contributions:** K.P. (Kanghyun Park), S.P. and B.Y. provided academic direction, conceptualization, validation, funding acquisition, writing—reviewing and editing. S.J. contributed especially in the synthesis and analysis of materials, wrote the introduction, results and discussion, experimental and conclusion parts. K.P. (Kyeongmun Park) contributed regarding adsorption of heavy metals and phosphate of HAP. S.S. and H.L. took part formal analysis such as XPS and antibacterial tests. All authors have read and agreed to the published version of the manuscript.

**Funding:** This research was supported by the Basic Science Research Program through a National Research Foundation of Korea (NRF) grant funded by the Korean Government (MSIP) (NRF-2020R1I1A3067208) and Korea Basic Science Institute (National research Facilities and Equipment Center) grant funded by the Ministry of Education (2021R1A6C101A429).

**Institutional Review Board Statement:** Not applicable.

**Informed Consent Statement:** Not applicable.

**Data Availability Statement:** Data are contained within the article.

**Conflicts of Interest:** The authors declare no conflict of interest.

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
