# Peer review of "Removal of Various Hazardous Materials Using a Multifunctional Biomass-Derived Hydroxyapatite (HAP) Catalyst and Its Antibacterial Effects"

_water, doi:10.3390/w13223302_

Round 1
Reviewer 1 Report
Comments for authors: Abstract: Standard deviations of the percentages are missing from the heavy metal removal rates and the antibacterial activity of HAP against E coli and S aureus. Introduction: Line 67: Missing references. Line 68: Missing references. Context unclear. Line 94: Mention the antibacterial efficiency (%) of t Results and Discussion: Line 201: Figure 4: the values on the x and y axis are difficult to read. Make the fonts larger. Line 214: Why was pH 8 not included for the experimentation? Line 227: Figure 5: the values of the side bars are difficult to read. Make it larger ¬¬ Line 244: Figure 6. The phosphate removal efficiency seems to have been carried out on a single sample. There are no error bars in the points plotted on the graph? Are we missing repeats? Line 265: Figure 7(a) are the antibacterial efficiency done in duplicates/triplicates? Error bars missing in the figure 7(a). Materials and Methods: Line 305: Earlier in line 214 the range of pH4-8 was mentioned but pH 8 has been left out. Justification for it? Line 335-336 : Good choice of including both gram +ve and gram -ve bacteria. However, the concentrations of HAP chosen seem random at best. Why were these 2 different concentrations of HAP chosen i.e. 5 and 10mM? The justification for the choice of concentration/s that not shown. Conclusion: Well summarized.Author Response
Reviewer 1
Comment 1: Standard deviations of the percentages are missing from the heavy metal removal rates and the antibacterial activity of HAP against E coli and S aureus.
Response: Thank you for your kind suggestions. In the accompanying information, we offer the RSD of the IPC values for heavy metals. However, RSD could not be represented in the case of antibacterial properties since the original UV value was the most accurate. We have re-drafted the manuscript provided. (Please see: Supporting information Table S1 and S2)
Comment 2: Line 67: Missing references. Line 68: Missing references. Context unclear. Line 94: Mention the antibacterial efficiency (%) of t.
Response: Thank you for your kind notice. We have re-drafted the manuscript provided. (Please see: Manuscript Page 2, Line 69 and Line 93-96)
Comment 3: Line 201: Figure 4: the values on the x and y axis are difficult to read. Make the fonts larger.
Response: Thank you for your kind notice. We have re-drafted the manuscript provided (Please see: Manuscript Page 7, Line 202).
Comment 4: Why was pH 8 not included for the experimentation?
Response: Thank you for your kind notice. If the pH of the aqueous solution is alkaline, the OH group interacts with Ca to generate a Ca(OH)2 precipitate. As a result, it cannot be evaluated under pH 8 or higher conditions.
Comment 5: Figure 5: the values of the side bars are difficult to read. Make it larger.
Response: We sincerely appreciate reviewer’s notice. We have re-drafted the manuscript provided (Please see: Manuscript Page 8, Figure 5).
Comment 6: Figure 6. The phosphate removal efficiency seems to have been carried out on a single sample. There are no error bars in the points plotted on the graph? Are we missing repeats?
Response: We sincerely appreciate reviewer’s kind and instructive comments on our work. We compared the HAP employed in this work for phosphate ion elimination with CaO from oyster shells, a precursor. The error bar, however, could not be presented since their reaction rate was extremely quick and adsorption happened immediately following the reaction.
Comment 7: Figure 7(a) are the antibacterial efficiency done in duplicates/triplicates? Error bars missing in the figure 7(a).
Response: We sincerely appreciate reviewer’s kind and instructive comments on our work. The antibacterial efficacy test was only performed once. This is due to the fact that the more the same experiment is performed, the more lethal the findings for external bacteria contamination might be acquired. In the literature, we do state that HAP exhibits antibacterial capabilities at the nanoscale (Please see: Manuscript Page 2, Line 69-71, Reference 38).
Comment 8: Line 305: Earlier in line 214 the range of pH4-8 was mentioned but pH 8 has been left out. Justification for it?
Response: Thank you for your kind notice on our work. The pH range of wastewater varies in general. If the pH of the aqueous solution is alkaline, the OH group interacts with Ca to generate a Ca(OH)2 precipitate.
Comment 9: Line 335-336 : Good choice of including both gram +ve and gram -ve bacteria. However, the concentrations of HAP chosen seem random at best. Why were these 2 different concentrations of HAP chosen i.e. 5 and 10mM? The justification for the choice of concentration/s that not shown.
Response: Thank you for your kind notice. We tried 1 mM and 3 mM concentrations of HAP but didn't achieve excellent results. Antibacterial activity, on the other hand, was detected at 5 mM of HAP. As a result, 10 mM of HAP was utilized to confirm antibacterial activity based on the amount of HAP. Moreover, other papers did not make comparisons based on antimicrobial substance concentrations, but instead reported on the observation of colonies.
Reviewer 2 Report
This manuscript deals with the preparation of an effective catalyst for the adsorption of heavy metals from aqueous solutions. In general, the presented manuscript can be a great addition to the journal as authors present an interesting material which has been fully characterized. I recommend the acceptance of this manuscript after consideration of some minor comments:
- Authors did not include an analysis of the effect of pollutant initial concentration. I consider this analysis quite important as it can provide information from the isotherms models like the maximum adsorption capacity (qmax). This qmax is a more general value that considers the amount of adsorbent vs. the adsorbate.
- In the abstract, please include qmax instead of the percentage of removal efficiency.
- Please include in Figure 4, 5 and 6 the corresponding experimental conditions: amount of adsorbent, pollutant initial concentration, pH, time, etc.
- Figure 5 and 7 are bit difficult to read. Please increse font sizes.
- How about the material's reuse?
- It would be nice to see a comparison of the antibacterial effect with other work. I suggest authors to include a Table with the efficiencies of other similar materials.
Author Response
Reviewer 2
This manuscript deals with the preparation of an effective catalyst for the adsorption of heavy metals from aqueous solutions. In general, the presented manuscript can be a great addition to the journal as authors present an interesting material which has been fully characterized. I recommend the acceptance of this manuscript after consideration of some minor comments:
Comment 1: Authors did not include an analysis of the effect of pollutant initial concentration. I consider this analysis quite important as it can provide information from the isotherms models like the maximum adsorption capacity (qmax). This qmax is a more general value that considers the amount of adsorbent vs. the adsorbate. In the abstract, please include qmax instead of the percentage of removal efficiency.
Response: Thanks for your kind suggestion and we fully agree with the reviewer’s comment. However, we used previous work, reference 19, in calculating removal efficiency. Figures 4b and 5d also demonstrate the adsorption capacity of each heavy metal. (Please see: Manuscript Page 6, Figure 4 and Page 7, Figure 5.)
Comment 2: Please include in Figure 4, 5 and 6 the corresponding experimental conditions: amount of adsorbent, pollutant initial concentration, pH, time, etc
Response: Thank you for your kind notice. We have re-drafted the manuscript provided. (Please see: Manuscript Page 6, Line 204-205, Manuscript Page 7, Line 232-234 and Line 250-251.)
Comment 3: Figure 5 and 7 are bit difficult to read. Please increase font sizes.
Response: We sincerely appreciate reviewer’s notice. We have re-drafted the manuscript provided (Please see: Manuscript Page 8, Figure 5 and Manuscript Page 9, Figure 7).
Comment 4: How about the material's reuse?
Response: We sincerely appreciate reviewer’s suggestion. In reality, the metal ion must undergo ion exchange in the hap structure. As a result, there are no reusable materials. And we have previously discussed the adsorption mechanism in equations 5, 6, 7, and 8. (Please see: Manuscript Page 5, Equation 5, 6, 7, and 8).
Comment 5: It would be nice to see a comparison of the antibacterial effect with other work. I suggest authors to include a Table with the efficiencies of other similar materials.
Response: Thank you for your kind suggestion. However, we attempted to discover references with findings of methods similar to the experimental settings in our work, but because many articles investigated colonies, comparing them proved impossible.